# Relationship of Stiffness-Based Indentation Properties Using Continuous-Stiffness-Measurement Method

**DOI:** 10.3390/ma13010097

**Published:** 2019-12-24

**Authors:** Wai Yeong Huen, Hyuk Lee, Vanissorn Vimonsatit, Priyan Mendis

**Affiliations:** 1Civil and Mechanical Engineering, Curtin University, 6102 Perth, Australia; lee.lee@curtin.edu.au (H.L.); v.vimonsatit@curtin.edu.au (V.V.); 2Infrastructure Engineering, University of Melbourne, 3010 Melbourne, Australia; pamendis@unimelb.edu.au

**Keywords:** nanoindentation, stiffness, elastic modulus, hardness.

## Abstract

The determination of elastic modulus (*E*) and hardness (*H*) relies on the accuracy of the contact area under the indenter tip, but this parameter cannot be explicitly measured during the nanoindentation process. This work presents a new approach that can derive the elastic modulus (*E*) and contact depth (*h_c_*) based on measured experiment stiffness using the continuous-stiffness-measurement (CSM) method. To achieve this, an inverse algorithm is proposed by incorporating a set of stiffness-based relationship functions that are derived from combining the dimensional analysis approach and computational simulation. This proposed solution considers both the sink-in and pile-up contact profiles; therefore, it provides a more accurate solution when compared to a conventional method that only considers the sink-in contact profile. While the proposed solution is sensitive to Poisson’s ratio (*ν*) and the equivalent indentation conical angle (*θ*), it is not affected by material plasticity, including yield strength (*σ_y_*) and work hardening (*n*) for the investigated range of 0.001 < *σ_y_/E* < 0.5. The proposed stiffness-based approach can be used to consistently derive elastic modulus and hardness by using stiffness and the load-and-unload curve measured by the continuous-stiffness-measurement (CSM) method.

## 1. Introduction

Nanoindentation is a commonly used technique to investigate submicron or nanoscale mechanical material properties such as elastic modulus (*E*) and hardness (*H*). These properties can be deduced from the load-unload curve using a popular method proposed by Oliver and Pharr [1,2], also known as the Oliver and Pharr method. Central to the Oliver and Pharr method is the determination of the projected area from the surface formed by the indenter’s impression, also referred to as the contact area. Determining the indentation’s impression area is a complex process where analytical solutions are difficult to obtain due to its dependence on the material’s elastic and plastic behaviour [3,4]. The original Oliver and Pharr method relies on the elastic-contact analytical solution first introduced by Sneddon [5], who described the shape of an indentation impression using a solid revolution of a nonlinear function. As a result, the Oliver and Pharr method generally apply well to material that has a sink-in contact profile, where the contact point between indenter and material is lower compared to that of the original material surface. However, for the same reason, it is unclear how well the Oliver and Pharr method can be used for material that exhibits a pile-up profile, where the contact area between indenter and material is higher than that of the original material surface. Both the sink-in and pile-up contact profiles are illustrated in Figure 1.

For this reason, researchers often cautioned the use of the Oliver and Pharr method when applied to a wide range of materials with contact profiles that no longer resemble the assumed elastic analytical solution [6,7,8].

To address this issue, other researchers attempted to improve accuracy in determining the contact area by visual methods [9,10,11], a semiempirical approach [12,13], and the computational finite-element approach [8,14,15]. These efforts were further extended to investigate the relationship of the material’s mechanical properties with the contact area using the dimensional-analysis approach [16,17,18,19]. The dimensional-analysis approach allows the derivation of the material’s mechanical properties such as elastic modulus (*E*), hardness (*H*), yield strength (*σ*_y_), and work-hardening exponent from a set of relationships using a specific algorithm to extract key information from a single nanoindentation load-unload curve [20,21,22]. Another key contribution is provided by the continuous-stiffness-measurement (CSM) method from the nanoindentation technique where direct measurement of the material’s stiffness is made possible. Dimensional functions created using measured stiffness with indentation force and depth has been shown to relate to the material’s elastic-plastic properties [23,24]. Solving these complex dimensional functions can be achieved by making an assumption about the contact condition using a reverse algorithm [21,24,25] or using an artificial-neural-network tool [26].

In this paper, a procedure to determine the contact area is presented by using a combination of the dimensional-analysis approach and CSM. This is achieved by relating the contact area to dimensional functions containing stiffness and other conventional nanoindentation parameters obtained from the load-unload curve. The determined contact area using the proposed relationship in this work is comparable to those derived from a conventional methodology [1,27]. The methodology proposed in this work is considered to be an improvement on the conventional Oliver and Pharr method, where curve fitting is required to estimate material stiffness. In the present work, curve fitting is no longer required as stiffness can be readily measured by the CSM method. Furthermore, the contact area derived using the dimensional-analysis approach does not depend on the shape of the indentation impression as prerequisite information. Hence, it can be applied to materials with both a sink-in and a pile-up contact profile.

## 2. Theoretical Analysis

The continuous-stiffness-measurement method using nanoindentation can continuously measure material stiffness along with indentation depth [1]. From the measured loading and unloading responses, stiffness can be related to indentation force and depth, as shown in Equations (1) and (2). The gradient, represented by stiffness–depth and stiffness–force coefficients (*C_h_* and *C_f_*), was observed to be linear for isotropic and homogeneous material [23]. Conventionally, the loading-curve gradient (*C*) can be defined as the gradient of the change in force (*F*) and the square of the displacement (*h*^2^). By substituting Equations (1) and (2) into Equation (3), *C* can be derived from the stiffness coefficients with *C = C_f_/C_h_*^2^, as shown in Equation (3).
(1)$$h=C_{h}\cdot S$$
(2)$$\;F=C_{f}\cdot S^{2}\;$$
(3)$$\;F=C\cdot h^{2}\;$$

These relationships can be further extended to the material’s elastic and plastic parameters, such as elastic modulus, yield strength, and work-hardening exponent using a dimensional-analysis approach [16,26]. Cheng and Cheng [28] proposed to relate indentation force (*F*) and stiffness (*S*) with mechanical properties using dimensional functions, as shown in Equations (4) and (5). Maximum stiffness (*S*|_*h=h_m_*_) referred to in Equation (5) is taken at the maximum indentation depth, where indention depth (*h*) is equal to maximum indentation depth (*h_m_*).
(4)$$\frac{F}{E\;h^{2}}=\frac{C}{E}={\;\Pi }_{1}\left(\frac{\sigma_{y}}{E},n,v,\theta \right)$$
(5)$$\;\frac{1}{Eh_{m}}{S|}_{h=h_{m}}=\Pi_{2}\left(\frac{\sigma_{y}}{E},n,v,\theta \right)\;$$

Conventionally, maximum stiffness at maximum indentation depth is calculated by finding the gradient of the unloading curve through curve fitting [1]. However, with the CSM method, the stiffness can be readily measured from the indentation process. From Equation (1), stiffness has a linear relationship with indentation depth, so Equation (4) can be rewritten as
(6)$$\frac{S}{Eh}=\frac{1}{C_{h}E}=\Pi_{3}\left(\frac{\sigma_{y}}{E},n,v,\theta \right).$$

Equation (5), initially proposed by Cheng and Cheng [28], is effectively the subset of the dimensional function in Equation (6), where stiffness (*S*) can be measured along with the entire indentation depth (*h*). Combining Equations (4) and (6) yields Equation (7), which introduces new dimensionless parameter *C_h_C*. Since both dimensionless parameters *C_h_C* and *C/E* are shown to relate to *σ_y_*/*E*, this paper proposes a new dimensionless function by directly relating *C_h_C* with *C/E*, as shown in Equation (8).
(7)$$C_{h}C=\Pi_{4}\left(\frac{\sigma_{y}}{E},n,v,\theta \right)$$
(8)$$\;\frac{C}{E}=f\left(C_{h}C\right)\;$$

Cheng and Cheng’s method further relates contact depth (*h_c_*) with mechanical properties using the dimensional-function approach [16], as shown in Equation (9).
(9)$$\frac{h_{c}}{h_{m}}=\Pi_{5}\left(\frac{\sigma_{y}}{E},n,v,\theta \right)$$

Combining Equations (6) and (9), a dimensionless function is proposed as seen in Equation (10), where contact depth (*h_c_*) can be solved given elastic modulus *E*, which can be derived using Equation (8).
(10)$$\frac{h_{c}}{h_{m}}=f\left(C_{h}E\right)$$

Once contact depth (*h_c_*) is established, contact area *A_c_* can be determined with Equation (11), where *θ* = 70.3°, which is the equivalent inclination angle for the Berkovich indenter used in the present nanoindentation test. Hardness (*H*) can be derived by dividing maximum indentation force (*F_max_*) with contact area (*A_c_*), as shown in Equation (12).
(11)$$A_{c}=\pi \;tan^{2}\theta \;h_{c}{}^{2}$$
(12)$$\;H=\frac{F_{max}}{A_{c}}\;$$

## 3. Finite-Element Simulation

Finite-element simulation is used to replicate the indentation process with a selected material’s elastic-plastic behaviours. The selected material parameters include elastic modulus (*E*), yield strength (*σ_y_*), work-hardening exponent (*n*), Poisson’s ratio (*ρ*), and the equivalent indenter (conical) tip’s angle (*θ*). In total, 2496 combination runs were executed in the simulation (see more details in Appendix A). Commercial finite-element software (ANSYS, Cannonsburg, PA, USA) was used to create the 2-dimensional axisymmetric elastic-plastic model of equal height and width of 10 µm, 4 or 6 nodes bilinear element (PLANE183), modelled with a rigid conical indenter (see Figure 2). The conical indenter is commonly treated as a representation of pyramidal indenters by adopting an equivalent cone angle so that both the simulated indenter and the actual experiment indenter has the same area to depth relationship [29]. For this reason, the equivalent angle of 70.3°, which corresponds to the Berkovich indenter, was used throughout the analysis carried out using the experimental test data in this work. The indentation simulation was carried out using displacement controlled up to a depth of 2 μm with nonlinear analysis. To simulate the CSM application, a relatively small static displacement of 1 nm at a 100 nm interval is applied along with the entire 2 μm indentation depth. Stiffness (*S*) was obtained in postprocessing by acquiring the corresponding forces (*dF*) at each 1 nm (*dh*) of static displacement, i.e., *S = dF/dh*. This simplification in replicating CSM processing was proven to correctly represent the target experimental dynamic oscillation of 45 Hz in a previous study [24].

## 4. Experiment Preparation

Three materials, namely, bulk aluminum, bulk steel, and fused silica samples, were positioned in an epoxy resin casting with an exposed surface. The epoxy resin casting with the samples have specific size and depth that will meet the Nano Indenter G200 (Keysight Technologies, Santa Rosa, CA, USA) requirements. The samples are polished following standard ASTM E3-11 to reduce surface roughness. The surface roughness of the samples was then checked using the nanovision method. Further polishing was carried out using diamond-particle suspension liquid supplied by Thermal Fischer Scientific, Malaga, Australia, if surface roughness exceeded the recommended limit. The indentation was carried out with a Berkovich indenter (Keysight Technologies) using the standard XP CSM method with a maximum indentation depth preset at 2 µm. A Berkovich indenter was used in this work because of its geometrical similarity and its ability to better dynamically correlate with stiffness measurement [29]. Pyramidal indenters generate constant strain that is independent of the applied load along with indentation depth. For this reason, hardness measurement using a pyramidal indenter is independent of load. Due to the geometrical similarity, the continuous-stiffness-measurement outcome can then be directly correlated with contact depth and other mechanical properties. Experiment data were analysed in this work using the proposed algorithm to determine the corresponding contact area. A total of 200 indentations were done using the continuous-stiffness-measurement method for all 3 materials.

## 5. Results and Discussion

For the equivalent cone angle of 70.3°, a relationship for *C/E* and *C*·*C_h_* was found with varying Poisson’s ratio (*ν*), as shown in Equation (13) and plotted in Figure 3. The relationship between *C/E* and *C*·*C_h_* was found to be independent of work hardening. We chose to use a power-law function to represent this relationship, which allows the incorporation of Poisson’s ratio (*ν*) as a parameter to improve estimation accuracy. With both known values of the loading coefficient (*C*) derived from Equation (3), and the stiffness–depth coefficient (*C_h_*) derived from Equation (13), Equation (13) can be obtained for the elastic modulus (*E*). Equation (13) is plotted in Figure 3, showing the variation of Poisson’s ratio.
(13)$$\frac{C}{E}=\left(7.65\nu^{2}-2.266\nu +3.747\right){\left(C\cdot C_{h}\right)}^{0.7314}$$

Once the elastic modulus (*E*) was determined, contact depth (*h_c_*) could be calculated from Equation (14) by substituting Poisson’s ratio (*ν*), maximum indentation depth (*h_m_*), and stiffness–depth coefficient (*C_h_*) derived from Equation (1). Results are shown in Figure 4, with varying Poisson’s ratios (*ν*).
(14)$$\frac{h_{c}}{h_{m}}=\left(-0.24\nu^{2}-0.0644\nu +0.172\right){\left(E\cdot C_{h}\right)}^{-0.9323}$$

To verify the derivation of the contact area using the methodology proposed in this work, the derived contact area was compared to an existing contact-area relationship proposed by Cheng and Cheng [27], shown in Equation (15), where *E* must be known as a prerequisite.
(15)$$\frac{1-\nu^{2}}{Eh_{c}\tan\theta }{S|}_{h=h_{m}}=2$$

For the compatibility of the range of materials used in the present case, Equation (15) was shown valid for an elastic-plastic solid with work hardening and residual stresses [28]. For materials that exhibit pile-up contact profiles, such as aluminum and steel, the Cheng and Cheng [27] method yields a higher contact depth in comparison with the method proposed in this work. Using the Oliver and Pharr method [1], contact depth can be calculated using Equation (16) where geometric correction factor *ϵ* = 0.75.
(16)$$h_{c}=h_{max}-\epsilon \frac{F_{max}}{{S|}_{h=h_{max}}}$$

Contact depth (*h_c_*) was determined using the Oliver and Pharr method and is compared and presented in Table 1. For material that exhibits a sink-in profile, such as fused silica, contact depth derived from both the methods is close.

The relatively high contact depth derived by the Cheng and Cheng method using Equation (15) may be explained by taking a closer look into the way this equation is derived. Equation (15) shows that the relationship of ${S|}_{h=h_{m}}\left[\left(1-\nu^{2}\right)/Eh_{c}\tan\theta \right]$ and *σ_y_/E* is constant when it is taken with a wider range of 0 < *σ_y_/E* < 0.5. However, a closer look reveals that the value of ${S|}_{h=h_{m}}\left[\left(1-\nu^{2}\right)/Eh_{c}\tan\theta \right]$ has a widespread when *σ_y_/E* < 0.1, as shown in Figure 5, for different equivalent cone angles (*θ*). To consider the data spread, a mean value was taken for the range of 0 < *σ_y_/E* < 0.5 and resulted in a higher value ranging from 2.1 to 2.4. Furthermore, by using the methodology proposed in this work, the calculated *σ_y_/E* was lower than 0.1 for both aluminum (0.0002) and steel (0.0006). Due to this, the deduced ${S|}_{h=h_{m}}\left[\left(1-\nu^{2}\right)/Eh_{c}\tan\theta \right]$ value proposed by Cheng and Cheng [27] would be even higher compared to the deduced mean value shown in Figure 5. Comparing the *h_c_/h_m_* ratio obtained in this work and those proposed by Cheng and Cheng [16], the higher value obtained using the Cheng and Cheng method should be reduced by adopting a higher mean ${S|}_{h=h_{m}}\left[\left(1-\nu^{2}\right)/Ea\right]$ value for *σ_y_/E* < 0.05. In the case of the equivalent cone angle *θ* = 70.3°, the reduction of the *h_c_/h_m_* ratio using the data in Figure 5 was estimated as 12%, which brings it closer to the *h_c_/h_m_* ratio determined in this work. With this adjustment, it further verifies that the proposed relationship presented in this work is in line with the values proposed by Cheng and Cheng [27].

A flowchart is presented in Figure 6 to illustrate the procedure for obtaining stiffness-based relationships. This proposed procedure is effectively a reverse algorithm to determine the respective elastic modulus and hardness based on a given load-unload curve. This proposed algorithm is only applicable for a Berkovich indenter with an equivalent cone angle of 70.3°.

The accuracy of the proposed stiffness-based relationship was compared to conventional methods, including the power-law method proposed by Oliver and Pharr [1,2] and Dao’s reverse analysis algorithms [21], using the dimensional-analysis approach. Results are presented in Table 2. The obtained elastic modulus and hardness results were between the derived values by the conventional Oliver and Pharr [1,2] and Dao’s method [21]. This agrees with the previous literature [7,8] that reported an underestimation of contact depth derived by the Oliver and Pharr method [1,2].

## 6. Conclusions

In this work, stiffness-based relationships for elastic modulus and contact depth were proposed. The proposed solution can consistently evaluate elastic modulus and contact depth for both sink-in and pile-up profiles using a single relationship function. Since stiffness is measured instead of a derived value, the proposed stiffness-based relationship can be used instead of following the conventional curve-fitting approach in the Oliver and Pharr method. The proposed relationships presented in this work offer a more accurate means to derive elastic modulus and hardness for material with a pile-up or sink-in contact profile compared to the conventional Oliver and Pharr method. The proposed stiffness-based relationships were not affected by a scatter in plasticity parameters including the yield strength and work hardening. For this reason, the proposed relationships were valid for the entire investigated rage of 0.001 < *σ_y_/E* < 0.5.

## Figures and Tables

**Figure 1 materials-13-00097-f001:**
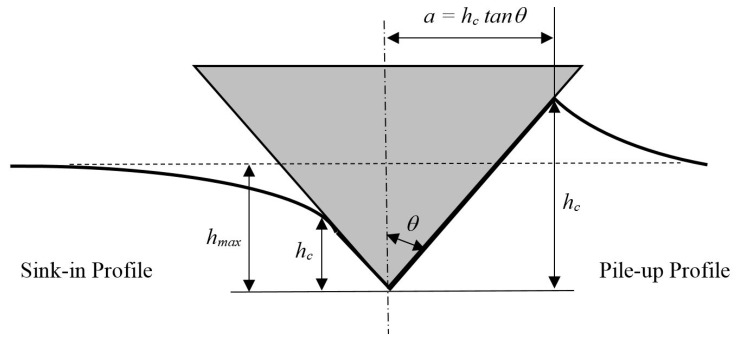
Sink-in and pile-up contact profile created on a sample during the nanoindentation process.

**Figure 2 materials-13-00097-f002:**
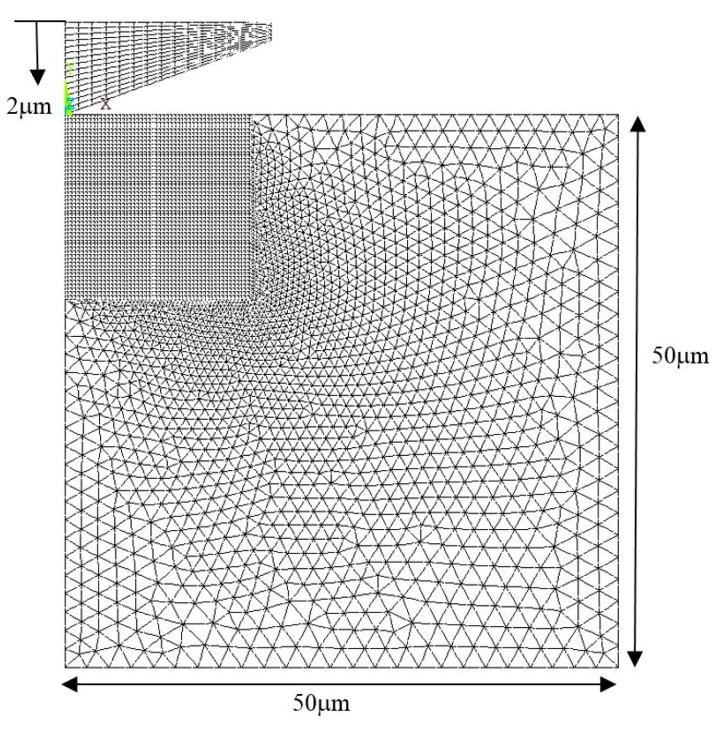
Nanoindentation finite-element model.

**Figure 3 materials-13-00097-f003:**
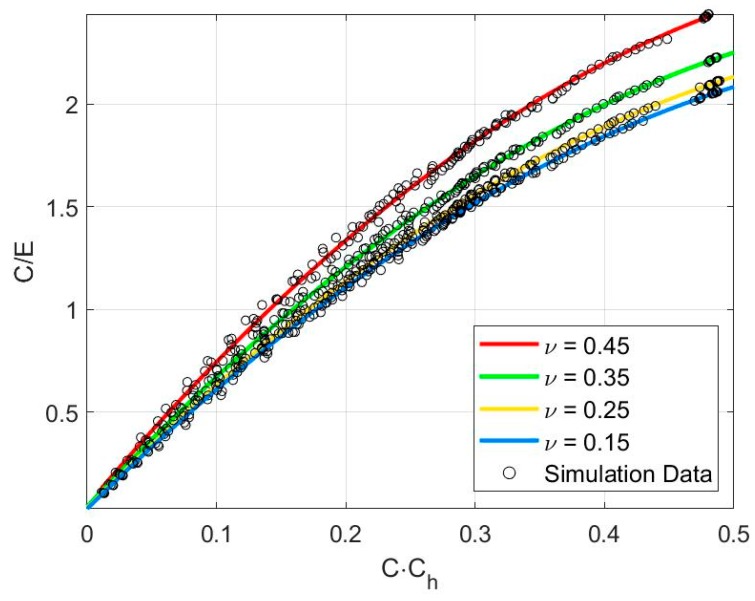
Dimensionless plot of *C/E* versus *C·C_h_* for *θ* = 70.3°.

**Figure 4 materials-13-00097-f004:**
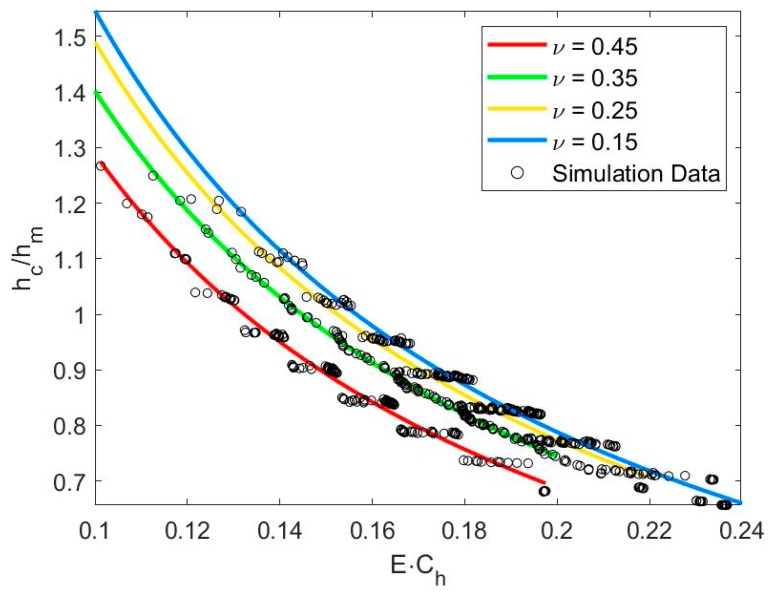
Relationship between *h_c_/h_m_* and *E*·*C_h_* for *θ* = 70.3°.

**Figure 5 materials-13-00097-f005:**
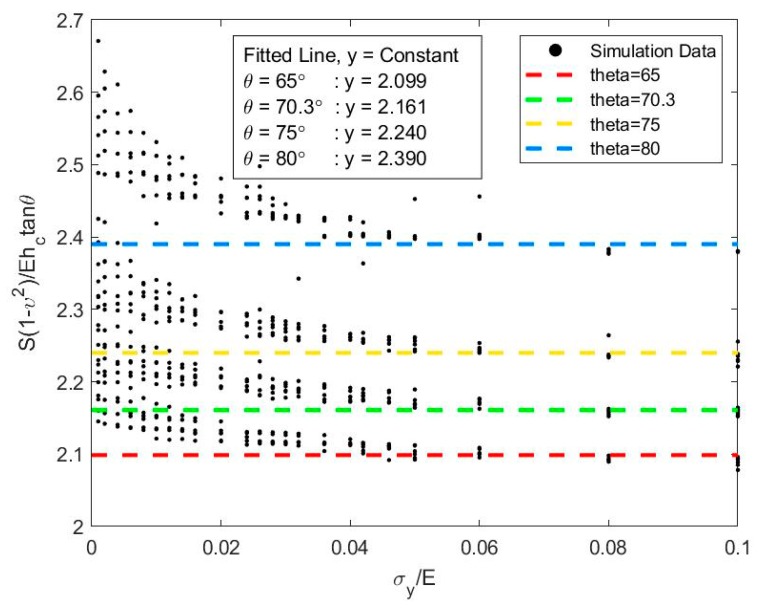
Conical indentation in elastic-plastic solid: relationship between ${S|}_{h=h_{m}}\left[\left(1-\nu^{2}\right)/Eh_{c}\tan\theta \right]$ and *σ_y_/E.*

**Figure 6 materials-13-00097-f006:**
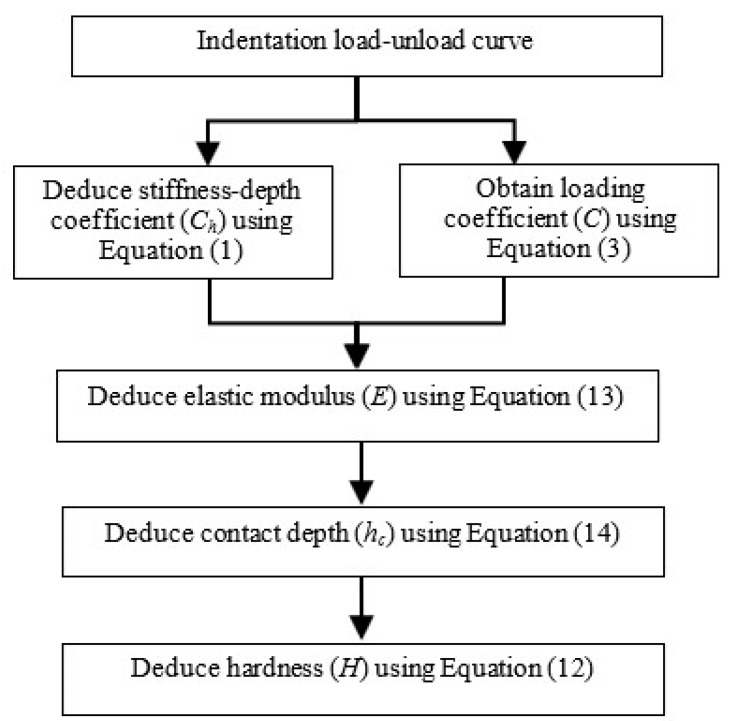
Stiffness-based reverse algorithm.

**Table 1 materials-13-00097-t001:** Contact depth *h_c_/h_m_* ratio comparison.

Materials	This Work (Equation (14))	Cheng and Cheng Method (Equation (15))	Oliver and Pharr Method (Equation (16))
Bulk aluminum	1.002	1.127	0.944
Bulk steel	1.083	1.438	0.974
Fused silica	0.732	0.772	0.690

**Table 2 materials-13-00097-t002:** Bulk aluminum properties using proposed stiffness-based solution and other conventional methods.

Indentation Solution	Elastic Modulus *E* (GPa)	Hardness *H* (GPa)
Stiffness-based relationship (this work)	87.4 ± 5.9	0.19 ± 0.019
Oliver and Pharr power-law method	110.3 ± 6.1	0.21 ± 0.017
Dao’s reverse-analysis algorithm	62.1 ± 10.7	0.07 ± 0.021

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
