# Peer review of "Relationship of Stiffness-Based Indentation Properties Using Continuous-Stiffness-Measurement Method"

_materials, 2019, doi:10.3390/ma13010097_

Round 1
Reviewer 1 Report
The paper is a valuable contribution to improving instrumented indentation as a technique for measuring the mechanical properties of materials. However, there are several opportunities for improvements:
(1) Since equations 1-10 are true for ideally sharp self-similar indenters (e.g., pyramids or cones), the authors should discuss whether the method they proposed in this paper, which is based on Eq. 1-10, is accurate for indenters with finite tip radii.
(2) Ref.[26] should be updated.
Reviewer 2 Report
Dear Authors and Editor,
In review, I received the paper entitled “Explicit Solution of Nanoindentation Contact Area by Continuous Stiffness Measurement Method” considered for publication in MDPI journal Materials. The article is focused on the problem of measuring the area under indenter during hardness and indentation modulus measurements. The topic addresses one important aspect of the evaluation of nanomaterials, and it is worth investigating. Regarding the text, I have the following comments:
Technical details and typos:
Introduction – Page 1/line 38, page 2/line 75, Page 3/line79 and elsewhere: (Error! Reference source not found…; call-outs for Equations and Figures). Fig 1 subtitles (Figure 1. Sink-in and pile-up contact profile Created on the sample during…), while the Image itself still has the text proof-reading markings (same goes for Fig. 6).
Despite the missing references and call-outs to Equations, I still managed to comprehend the text, but this has really (!) be solved prior to publication. Now regarding the more scientific part of the paper, I really appreciate the simulation work done, and the new (improved) method suggested. The paper would significantly gain on the impact if the simulation data could be compared by real experiments, but I assume this would be out of the scope of the current manuscript. I would suggest a bit extended explanation of Figure 4, about the unusually horizontal plots simulation data compared to v=45 and v=15, while not in case of v=35.
I suggest to the Editor and Author to resolve the technical issues and resubmit the paper.
Best regards, Reviewer.
Reviewer 3 Report
Review of the article: Explicit Solution of Nanoindentation Contact Area by Continuous Stiffness Measurement Method. Manuscript ref. number: materials-658285.
All sections: the in-text referencing seems to have an issue since in many cases the hypertext seems to be replaced by an error message (specifically for referencing equations and figures). This error usually happens during converting to PDF. Please consider a revision. Abstract: line 7 (and other places): Why are the authors using “E*” for the modulus? This nomenclature is usually used for complex modulus in viscoelasticity. The indentation modulus can be represented by “E.” Abstract: line 9 (and other places): What do the authors mean by explicit solution? An explicit solution is usually referred to a closed-form solution. While the solution incorporates an FE part, it cannot be considered as an explicit solution. Please correct. Introduction: Figure1: the red underlines in the figure need to be removed. Theoretical analysis: line 102 (and other places): This review wonders why the authors keep using the term “contact height?” Do they mean “contact depth?” If that is the case, please correct the manuscript. Finite Element Modeling: line 114: Please change the title of the section to “Finite Element Simulation.” Modeling is a totally different concept from Simulation. Finite Element Modeling: figure 2: What is the geometry of the head of the tip in simulation? Usually the tip is not perfect and has a blunt head which can be simulated as a spherical region with a small (~20 nm) radius. In addition, a sharp head would cause singularity in simulation. Finite Element Modeling: Why is the tip simulated as a rigid part? This might not be correct especially for indenting a material like steel. Experimental Preparation: This section is too short and not informative. Please add the details of the tests performed for this study.
Major Issues:
The authors seem to use the results of the CSM indentation to back-calculate the contact depth (or area). However, it seems that they don’t consider the fact that the modulus measured during CSM indentation (which is an input for their model) is calculated using the contact area based on contact depth and tip area calibration (see, e.g., equation 2-4 in ref. [1] below). So, they are calculating a parameter the value of which is already assumed during the measurement. This fact undermines the basis of this manuscript unless the authors provide acceptable justification. Why are the authors calibrating their model based on the results for steel and aluminum? These material show considerable indentation size effect that seems not to be considered in the model.
References:
[1] Voyiadjis, George Z., Leila Malekmotiei, and Aref Samadi-Dooki. "Indentation size effect in amorphous polymers based on shear transformation mediated plasticity." Polymer 137 (2018): 72-81.
Round 2
Reviewer 2 Report
The authors properly addressed all issued in the first review. I have no more objections prior to publication.
Reviewer 3 Report
The revised version can be published as is.